# The Role of the *AGPAT2* Gene in Adipose Tissue Biology and Congenital Generalized Lipodystrophy Pathophysiology

**DOI:** 10.3390/ijms26115416

**Published:** 2025-06-05

**Authors:** Maria Eduarda Cardoso de Melo, Letícia Marques Gomes da Silva, Ana Carolina Costa Cavalcante, Josivan Gomes Lima, Julliane Tamara Araújo de Melo Campos

**Affiliations:** 1Laboratório de Biologia Molecular e Genômica, Departamento de Biologia Celular e Genética, Centro de Biociências, Universidade Federal do Rio Grande do Norte (UFRN), Natal 59072-900, RN, Brazil; eduarda.melo.089@ufrn.edu.br (M.E.C.d.M.); leticia.silva.017@ufrn.edu.br (L.M.G.d.S.); ana.cavalcante.120@ufrn.edu.br (A.C.C.C.); 2Laboratório de Genética Molecular e Metabolismo, Instituto de Medicina Tropical do Rio Grande do Norte, Centro de Biociências, Universidade Federal do Rio Grande do Norte (UFRN), Natal 59072-900, RN, Brazil; josivanlima@gmail.com; 3Departamento de Medicina Clínica, Hospital Universitário Onofre Lopes (HUOL), Universidade Federal do Rio Grande do Norte (UFRN), Natal 59012-300, RN, Brazil

**Keywords:** white adipose tissue, adipogenesis, genetic lipodystrophy, lipid metabolism

## Abstract

1-Acylglycerol-3-phosphate O-acyltransferase (1-AGPAT) is an enzyme family composed of 11 isoforms. Notably, 1-AGPAT 2, the most studied isoform since its discovery, is a critical enzyme in the triglyceride synthesis pathway, converting lysophosphatidic acid to phosphatidic acid. In addition, *AGPAT2* gene expression is shown to be essential for adipocyte development and maturation. Defects in *AGPAT2* are responsible for significant pathophysiological alterations related to adipose tissue (AT). Pathogenic variants in this gene are the molecular etiology of Congenital Generalized Lipodystrophy type 1 (CGL1), in which fatty tissue is absent from birth. Metabolically, these individuals have several metabolic complications, including hypoleptinemia, hypoadiponectinemia, hyperglycemia, and hypertriglyceridemia. Furthermore, numerous *AGPAT2* pathogenic variants that enormously affect the amino acid sequence, the tertiary structure of 1-AGPAT 2, and their transmembrane and functional domains were found in CGL1 patients. However, studies investigating the genotype–phenotype relationship in this disease are scarce. Here, we used bioinformatics tools to verify the effect of the main pathogenic variants reported in the *AGPAT2* gene: c.366-588del, c.589-2A>G, c.646A>T, c.570C>A, c.369-372delGCTC, c.202C>T, c.514G>A, and c.144C>A in the 1-AGPAT 2 membrane topology. We also correlated the phenotype of CGL1 subjects harboring these variants to understand the genotype–phenotype relationship. We provided an integrative view of clinical, genetic, and metabolic features from CGL1 individuals, helping to understand the role of 1-AGPAT 2 in the pathogenesis of this rare disease. Data reviewed here highlight the importance of new molecular studies to improve our knowledge concerning clinical and genetic heterogeneity in CGL1.

## 1. Introduction

The 1-AGPATs are a family of enzymes consisting of 11 isoforms, each encoded by a different gene [1,2,3,4,5,6,7,8,9], which present a similar function: they act as intermediate enzymes in the triacylglycerol (TAG) and glycerophospholipid (GPL) biosynthesis pathways [10,11]. These enzymes acylate 1-acylglycerol-3-phosphate (lysophosphatidic acid; LPA) at carbon-2 (sn-2) to produce 1,2-diacylglycerol-3-phosphate (phosphatidic acid; PA) [12,13,14]. Despite this general feature, several AGPATs can also esterify lysophospholipids with polar heads, such as choline, serine, inositol, ethanolamine, and glycerol [15].

Among the enzymes of this family, 1-AGPAT 2 has been the most extensively studied since its discovery. This enzyme is highlighted by its remarkable and essential role in providing PA for the synthesis of TAG or GPL. It is abundant in AT but presents lower expression in the liver and pancreas [16,17,18,19]. According to the Human Protein Atlas, *AGPAT2* mRNA has high levels in the liver, AT, breast, and pancreas, while the 1-AGPAT 2 protein is abundant in the small intestine, bone marrow, breast, cerebral cortex, liver, AT, and other tissues [20], highlighting the functionality of the protein in homeostasis.

Moreover, due to its indispensable functions in adipocytes, genetic defects in this enzyme result in Congenital Generalized Lipodystrophy type 1 (CGL1—OMIM #608594) [21]. Thus, adipogenesis regulation becomes a predisposing factor for disorders related to dysfunctional AT formation and maintenance [22]. For the good functioning and structuring of this tissue, the pathway for TAG and GLP biosynthesis becomes essential.

The *AGPAT2* gene contains six exons and encodes the 1-AGPAT 2 protein. Homozygous or compound heterozygous pathogenic variants in this gene result in CGL1, a rare autosomal recessive disease. The scarcity of body AT since birth in CGL1 individuals results in the progression of severe metabolic disorders, usually before puberty. These metabolic derangements include hyperinsulinemia, insulin resistance, type 2 diabetes mellitus, and the early onset of hepatic steatosis [23]. Nevertheless, these metabolic disturbances resemble those observed in patients with generalized or central obesity [24]. Furthermore, a recent study reported the occurrence of pathogenic variants in heterozygosity in the *AGPAT2* gene. However, the patient’s clinical phenotype was related to partial lipodystrophy, characterized by a significant loss of AT in the extremities and low leptin levels, which in turn resulted in a remarkable preservation of functional AT [25,26]. The worldwide CGL prevalence ranges from 0.2 to 1 case per 1 million inhabitants [27]. The estimated CGL prevalence in Brazil is 3.23 per 100 thousand inhabitants [28]. Lima et al. identified the main causes of death and found that the mean age of death of CGL individuals in Brazil was 27.1 ± 12.4 years [29]. The current treatment for CGL is metreleptin, an analog of leptin, which is an adipokine produced by AT [30].

It is worth noting that associations of the *AGPAT2* gene with the development of cancers have been described [31]. It is first known that the 1-AGPAT 2 enzyme participates in the formation of TAG and GPL, which are essential in several aspects, such as integrating the formation of lipid droplets (LDs) in adipocytes and participating in cell signaling [13]. From this perspective, cancer cells require numerous resources to proliferate, including nucleic acids, proteins, and lipids, which prompts them to alter the conformation of their metabolism to store more precursors of these resources [32]. Thus, fatty acid sets originate from either exogenous sources or de novo fatty acid synthesis; the latter route is preferred by tumors, whereas normally functioning human cells utilize more exogenous sources [33]. Given these factors, it has been possible to correlate the participation of the *AGPAT2* gene in malignancies since 1-AGPAT 2 is an indispensable enzyme for lipid metabolism, which is necessary for cancer development. However, the role of *AGPAT2* in cancer is out of the scope of this review. A timeline concerning the main discoveries related to the *AGPAT2* gene and AT biology is provided in Figure 1.

We have gathered the current literature on the *AGPAT2* gene to provide an overview of its diverse roles. The data presented in this article were obtained from a literature search conducted in the PubMed database of the National Center for Biotechnology Information (NCBI). The articles included in this review were dated until the end of June 2024. “*AGPAT2*” was used as a search word, resulting in 171 articles at the time of writing this article. From there, papers were screened based on the importance of *AGPAT2* to that study. Fifty-seven articles were excluded after reviewing the title and abstract, as they addressed subjects unrelated to this paper. Another 18 articles were excluded because they focused on a different lipodystrophy, in general. Finally, 15 articles that only cited *AGPAT2*, without an association with adipogenesis, were also excluded. Thus, 81 articles were included in this review. A flowchart of this review is included in Figure 2.

To better understand the genotype–phenotype correlation associated with the *AGPAT2* gene and CGL1 and enlarge this review with new data, we performed the membrane topology predictions for the most frequent *AGPAT2* pathogenic variants: c.517-588del, c.589-2A>G, c.646A>T, c.570C>A, c.369-372delGCTC, c.202C>T, c.514G>A, and c.144C>A (Table 1). Further, we collected clinical, phenotypic, and genetic data from CGL1 subjects harboring these *AGPAT2* pathogenic variants. The *AGPAT2* genotype–phenotype relationship is provided in Table 2. The genomic sequence of the *AGPAT2* gene (GRCh38 from Genome Reference Consortium Human) was obtained from the NCBI database using the Genome Browser. The sequences for all the aforementioned variants were manually obtained according to the variant being analyzed. Next, these sequences were analyzed using the following software: PSIPRED—MEMSAT-SVM (Membrane Helix Prediction, available at http://globin.bio.warwick.ac.uk/psipred/; accessed on 3 March 2024), SOSUI (available at http://www.tuat.ac.jp/mitaku/sosui/; accessed on 3 March 2024), TMHMM (available at http://www.cbs.dtu.dk/services/TMHMM/; accessed on 4 March 2024), and T-COFFEE (available at http://tcoffee.crg.cat; accessed on 4 March 2024) [34,35,36,37]. All software predictions are based on the secondary structure of wild-type 1-AGPAT 2 and mutated 1-AGPAT 2 proteins. Since some *AGPAT2* pathogenic variants were named before the establishment of Human Genome Variation Society (HGVS) recommendations, all included variants reviewed here were classified according to HGVS guidelines, and the pathogenicity was confirmed according to the American College of Medical Genetics and Genomics (ACMG) criteria [38,39]. The Mutalyzer tool version 3.1.1 (available at: https://mutalyzer.nl accessed on 6 April 2024) was used to confirm the HGVS nomenclature [40].

Throughout this review, we have highlighted the recent studies that shed light on the functions and regulatory mechanisms of *AGPAT2*. It is notorious from the available evidence that *AGPAT2* plays a critical role in various physiological processes, including lipid metabolism, adipogenesis, and cell signaling. Additionally, emerging research has also linked *AGPAT2* to different pathological conditions, such as obesity, insulin resistance, and cancer. The comprehensive understanding of *AGPAT’s* involvement in these processes provides a foundation for future investigations that could unveil novel therapeutic targets. Overall, this review contributes to the current knowledge of *AGPAT2* and emphasizes the need for further research to comprehend its multifaceted functions and implications in human health and disease.

## 2. The 1-AGPAT 2 Protein and Its Motifs

AGPATs correspond to an enzyme family comprising 11 isoforms [4,8,21,41,42,43]. AGPATs act by converting LPA to PA, a precursor for synthesizing phospholipids (PLs) and diacylglycerol (DAG), molecules with essential roles in signal transduction and lipid biosynthesis [14,44]. The family of AGPATs exhibits significant identity among its proteins and shares extensive sequence similarities with microbial, plant, and animal AGPATs [14]. In this family, two major motifs are highly conserved among the proteins: in the amino-terminal region and EGTR in the middle part of the protein, which are responsible for its catalytic activity [13,22]. EGTR is also a domain related to substrate recognition and binding [45]. The NHX4D motif is conserved in all AGPATs of the family, which is not the case for the EGTR domain [4,7,8]. Two other motifs have also been described for 1-AGPAT 2: FINR, between amino acid residues 143 and 149, and IVPV from amino acid (aa) residues 205 to 208 of the protein [46,47]. 1-AGPAT 2 is the best-studied protein of this family at the physiological level. It is encoded by the *AGPAT2* gene, which is located on chromosome 9 (9q34.3). The 1-AGPAT 2 protein has a molecular weight of 31 kDa and presents two main isoforms: the shortest has 246 aa, and the longest has 278 aa [21]. The longest 1-AGPAT 2 protein is highlighted in Figure 3. 1-AGPAT 2 is localized on the endoplasmic reticulum (ER) and has four transmembrane domains. Its mRNA exhibits higher expression levels in the AT, heart, and liver [14], and its genomic sequence comprises six exons within a genomic region spanning 11.4 kb [45]. An analysis of murine 1-AGPAT 2 demonstrated that the polypeptide contains a putative signal sequence in its N-terminal region with a predicted cleavage site between residues 45 and 46 [20]. Studies conducted with the longest 1-AGPAT 2 isoform indicated the presence of four potential hydrophobic regions. Among them, three regions were identified as potential transmembrane helices: two at the N-terminal, between positions 4 and 50, and one between residues 122 and 143 [45]. In addition to the well-conserved motifs, the C-terminal residues are essential determinants of 1-AGPAT 2 enzymatic activity [48]. A study demonstrated a novel conserved KX_2_LX_6_GX_12_R motif/pattern found in murine AGPATs between the catalytic and substrate-binding motifs. However, this was the only study to describe this motif [45]. More studies are crucial to scrutinize the impact of each domain on protein–protein interactions between 1-AGPAT 2 and other enzymes related to adipogenesis.

## 3. The Role of 1-AGPAT 2 in the Biosynthesis of Triacylglycerols

Most fatty acids that are synthesized or ingested have two main destinations: the formation of TAGs in the AT for energy storage or the generation of membrane PL. Both pathways start from the formation of glycerol acyl fatty esters. Glycerol-3-phosphate is converted into LPA, which will subsequently be converted to PA by AGPATs. 1-AGPAT 2, the most studied isoform of AGPATs, esterifies the sn-2 carbon of LPA [15,24]. The PA resulting from this reaction could be used to form DAGs and, subsequently, TAGs [49] (Figure 4).

In this sense, LPAs are signal transduction molecules that interact with G protein-coupled receptors, induce adipocyte proliferation and fibronectin matrix assembly in fibroblasts, and protect T cells from apoptosis [50,51]. PAs are lipid second messengers participating in various intracellular signaling events and regulating a growing list of signaling proteins, including protein kinases and phosphatases [52]. PAs and DAGs also serve as intermediates for the biosynthesis of GPLs, such as phosphatidylcholine (PtdCho), phosphatidylserine (PS), phosphatidyl inositol (PtdIns), cardiolipin, and phosphatidylethanolamine (PtdEtn). These PLs are integral components of all cell membranes [53]. In this way, 1-AGPAT 2 is a crucial enzyme involved in the biosynthesis of TAGs and PLs. Therefore, the discovery of pathogenic variants in the *AGPAT2* gene has increased interest in these biochemical pathways and their implications in human physiology [54].

Defects in the *AGPAT2* gene are responsible for significant pathophysiological alterations related to AT formation and function [55,56,57,58,59,60]. The decrease or loss of 1-AGPAT 2 catalytic activity alters the conversion of LPA to PA and, consequently, the synthesis of several classes of lipids. In this way, it was initially believed that the almost complete absence of 1-AGPAT 2 activity would decrease the bioavailability of PA in adipocytes and cause an increase in LPA, raising suspicions about the activity of other isoforms in the conversion of this molecule [13]. Similarly, studies on C2C12 myoblast cells showed that the overexpression of 1- AGPAT 1, an enzyme in the same family as 1-AGPAT 2, was not associated with increased PA levels [61]. A study in wild-type mice overexpressing the human *AGPAT2* revealed increased subcutaneous white adipose tissue (sWAT), gonadal white adipose tissue (gWAT), and brown adipose tissue (BAT). However, no changes were observed in PL and TAG concentrations [62]. This last finding differs from that observed in 3T3-L1 preadipocytes, which presented increased TAG synthesis after *Agpat2* overexpression [63]. Subsequent studies in male *Agpat2*^−/−^ mice compared with wild-type mice demonstrated that hepatic levels of LPA and PA increased approximately 2-fold and 5-fold, respectively [64]. Furthermore, high levels of diacylglycerol kinase and phospholipase D were detected, showing an alternative route for synthesizing PAs in the liver [64]. New findings observed in mouse embryonic fibroblasts (MEFs) also corroborated the finding of increased PAs [65]. A recent study demonstrated decreased *Agpat2* levels in white adipose tissue (WAT) of adult mice, resulting in lipodystrophy and inflammation in both WAT and the liver. Furthermore, LPA is a crucial mediator of inflammation in the WAT and liver of rat models of CGL1 and overnutrition [66].

1-AGPAT 2 has a higher affinity for arachidonic acid as a substrate [13]. This raised the hypothesis that other AGPAT isoforms could synthesize these GPLs other than 1-AGPAT 2, and their fatty acid composition could be altered, affecting membrane organization, protein symmetry, functions, and orientation [53]. In this regard, later studies demonstrated PA as a critical intermediate in the biosynthesis of TAG and several specific PLs, such as PtdIns, PtdCho, PtdEtn, and cardiolipin, corroborating the hypothesis raised two years earlier [48]. However, unexpectedly, another research group reported that levels of several phospholipid species, including PA, are elevated in *Agpat2*^−/−^ adipocytes with TAG depletion, demonstrating that mutant 1-AGPAT 2 impairs PA availability for TAG synthesis but not for overall PA synthesis, nor its use in the synthesis of other PLs [67]. Furthermore, a recent study in a Chinese crab animal model (*Eriocheir sinensis*) demonstrated that the expression pattern of AGPAT isoforms indicates different functions during TAG synthesis, reflecting the importance of other isoforms in the AGPAT family [68]. Together, these data highlight the role of 1-AGPAT 2 in lipid metabolism and underscore the importance of investigating its enzymatic activity in the context of CGL1.

## 4. The Role of 1-AGPAT 2 in Adipogenesis

AT is widely recognized for its role in triglyceride storage and as a protective mechanism against mechanical impacts. Furthermore, this tissue also has an essential role in endocrine function. Adipogenesis is the process by which fully mature adipocytes develop from mesenchymal stem cells. The two main ATs include WAT and BAT, which have different structures and biological functions [69]. White adipocytes have a single large LD occupying most of the cell volume, with few mitochondria, displacing the nucleus peripherally [70]. Brown adipocytes are polygonal cells containing numerous small LDs (referred to as multilocular AT), with a central nucleus surrounded by a clear cytoplasm and numerous mitochondria [71,72]. The WAT accounts for most of the AT in adult humans and presents high plasticity. It is a critical site for energy homeostasis, insulin signaling, and endocrine action, secreting adipokines such as leptin and adiponectin, which are essential for body homeostasis [73]. BAT is predominantly responsible for thermogenesis and is mainly found in newborns and hibernating mammals [74]. Brown adipocytes have the potential to be metabolically beneficial, especially for obese individuals, as they have the potential to increase energy expenditure upon proper stimulation [75]. BAT has even been reported to prevent glucose intolerance and cardiac remodeling in mice on a high-lipid diet after a mild myocardial infarction [76]. Another AT type contains beige adipocytes, which are present within WAT and, when stimulated, acquire a brown fat phenotype, leading to increased thermogenesis [77]. This phenomenon is known as browning and is most likely to occur in subcutaneous fat deposits. The dynamism of this process is being studied for its potential use in combating various diseases. A recent study using *Agpat2*-null mice demonstrated that AT-specific re-expression of *Agpat2* resulted in partial regeneration of WAT and BAT (approximately 30–50% compared to wild-type mice), whereas silencing *Agpat2* expression caused a total loss of these AT tissues [78]. WAT and BAT adipocytes originate from distinct precursor cells. Adipoblast cells are generated from multipotent stem cells and are committed to differentiation into adipocytes. After forming preadipocytes, they undergo clonal expansion and, in response to specific stimuli, become mature adipocytes [70]. Mesenchymal stem cells from certain stimuli differentiate into WAT and beige AT cells, while Myf5 mesenchymal cells are responsible for the adipogenesis of BAT adipocytes [79]. Adipogenesis is a complex sequential process regulated in several ways. Initially, in WAT adipocytes, the activator protein-1 (AP-1) transcription factor complex (AP-1/C-FOS and AP-1/Fra-2) is activated at the early and late stages of adipogenesis, respectively [80]. This activation subsequently activates the central adipogenesis regulator, PPARγ (peroxisome proliferator-activated receptor gamma) [81,82]. Other factors such as STATs [83,84,85], sterol response element-binding protein-1 (SREBP-1) and C/EBP family members, insulin, IGF-1 (insulin-like growth factor), and BMPs also act as positive effectors and stimulators of this process [86]. In addition, certain factors that inhibit adipogenesis include proteins from the WNT signaling pathway and transforming growth factor beta (TGFβ) [87,88].

Several studies have sought to elucidate the role of the *AGPAT2* gene in the adipogenic process, as pathogenic variants in this gene are associated with a rare genetic disorder characterized by an extreme AT deficiency from birth. Gale et al. (2006) revealed that *Agpat2* knockdown in murine preadipocytes (OP9 cells) prevented the induction of transcriptional activators of adipogenesis, such as C/EBPβ and PPARγ, demonstrating the role of 1-AGPAT 2 in mediating the adipogenic pathway. Furthermore, this same study demonstrated the induction of *Agpat2* expression in OP9 cells differentiated into adipocytes [67]. These findings corroborate data from another study that demonstrated a novel interaction between the *CEBPA* and *AGPAT2* genes, specifically in the context of *C/EBPα-*dependent transcription of *AGPAT2* in human adipocytes [89].

Subauste et al. used two different models to verify the role of 1-AGPAT 2 in adipogenesis. They found that muscle-derived multipotent cells (MDMCs) isolated from *vastus lateralis* biopsies of CGL1 individuals displayed compromised adipogenesis. Similar results were found in 3T3-L1 preadipocytes after *Agpat2* knockdown. The lack of 1-AGPAT 2 activity reduced Akt protein activation, while constitutive overexpression of Akt partially restored lipogenesis. In other words, 1-AGPAT 2 regulates adipogenesis earlier by modulating the phosphatidylinositol-3kinase (PI3K)/Akt pathways [90].

Another study showed an interaction between 1-AGPAT 2 and seipin during adipogenesis, leading to the nuclear accumulation of PPARγ. Furthermore, the ER-localized 1 AGPAT-2 and lipin 1 proteins can directly interact with the ER-localized seipin protein, which is related to Congenital Generalized Lipodystrophy type 2 (CGL2) [91]. Furthermore, another study from the same research group demonstrated that seipin can interact with GPAT3 and simultaneously bind to GPAT3 and 1-AGPAT 2 to promote adipogenesis. Furthermore, the expression inhibition of these three proteins can impair the induction of early markers of adipogenesis in cultured adipocytes [92]. As reviewed by Qi et al. [93], PA formed from 1-AGPAT 2 activity is critical for LD growth and adipocyte development, highlighting the distinct roles of *AGPAT2* in AT biology.

Fernández-Galilea et al. found that interscapular BAT preadipocytes isolated from *Agpat2*^−/−^ newborn mice and cultured/differentiated preadipocytes required 1-AGPAT 2 for adipogenesis. Adipocytes lacking 1-AGPAT 2 showed fewer lipids and lower levels of adipocytic markers compared to cells expressing the *Agpat2* gene [94]. Furthermore, no evidence of increased caspase activation, autophagy, or apoptosis was found in *Agpat2-*lacking cells, indicating that these pathways are unrelated to the role of *Agpat2* in adipogenesis [94].

Cautivo et al. [65] found that adipocytes differentiated from *Agpat2*^−/−^ MEFs had impaired adipogenesis and ultrastructural abnormalities in LD, mitochondria, and the plasma membrane. However, PPARγ overexpression increased the differentiation of *Agpat2*^−/−^ MEFs into adipocytes; however, it did not prevent morphological abnormalities and cell death. Furthermore, they found that newborn *Agpat2*^−/−^ mice were deficient in caveolae and displayed abnormal LDs and mitochondria. Further, increased lipid accumulation in *Agpat2*^−/−^ mice liver coincided with AT degeneration.

Tapia et al. found that differentiated *Agpat2*^−/−^ brown adipocytes from mice had fewer lipid-laden cells, abnormal LDs, and reduced abundance of *Ppar*γ, *Pparα* (peroxisome proliferator-activated receptor alpha), *C/ebpα* (CCAAT enhancer-binding protein alpha), and *Pgc1α* (peroxisome proliferator-activated receptor gamma coactivator 1-alpha), both at the mRNA and protein levels when compared with wild-type cells [95]. They also found that 1-AGPAT 2 is required for normal BAT differentiation. Differentiated *Agpat2*^−/−^ brown adipocytes presented a lower proportion of lipid-laden cells, increased interferon-stimulated gene expression, changes in mitochondrial morphology and mass, and fewer mitochondria–LD contact points. Another study corroborating these findings revealed that *Agpat2*-deficient adipocytes had impaired adipogenesis and fewer caveolae while maintaining insulin signaling [96]. Taken together, these data highlight the role of the *Agpat2* gene in WAT and BAT adipogenesis.

## 5. *AGPAT2* and Lipodystrophy

Since the initial studies about the role of the *AGPAT2* gene, a striking fact was its high expression in AT. Therefore, the effect of decreasing its expression in AT began to be analyzed, and it was found that defects in 1-AGPAT 2 can lead to a significant abnormality in adipocyte differentiation and proliferation [65]. Moreover, the interruption of its expression can result in the development of a markedly reduced number of adipocytes. In Rio de Janeiro, Brazil, in 1954, Waldemar Berardinelli was the first to report Congenital Generalized Lipodystrophy (CGL), a condition caused by a disturbance in the biosynthetic pathway of TAGs [13,97]. Seip was the second researcher to report patients with Berardinelli–Seip syndrome [98]. However, the molecular mechanism leading to this pathophysiology remained uncertain. Initially, it was believed that the reduction in 1-AGPAT 2 enzyme activity underlies the loss of AT in individuals with CGL [48], without ruling out the possibility that the lack of AT could result from the inability to synthesize TAGs, PL, or PA, or even from the excessive accumulation of LPA. To elucidate this question, *Agpat2* knockdown in preadipocytes demonstrated that its activity is required for TAG mass accumulation in mature adipocytes. Furthermore, the same study found that *Agpat2* mRNA expression increased by approximately 30-fold during adipocyte differentiation [67]. Subsequently, it was demonstrated that fat transplantation in lipoatrophic mice with clinical symptomatology of CGL allowed for the reversal of diabetes [99,100,101], and the metabolic disturbances found in lipodystrophic mice and humans could be attenuated by administering leptin [102,103,104,105]. Assays in *Agpat2* knockout mice demonstrated the development of severe lipodystrophy affecting both white and brown AT and a metabolic condition characteristic of CGL. Further, in the same study, the mRNA and protein levels of 1-AGPAT 1 were markedly increased in the liver of *Agpat2^−/−^* mice, suggesting that the alternative monoacylglycerol pathway for triglyceride biosynthesis is activated in the absence of *Agpat2* [106]. A recent study in *Agpat2*^−/−^ mice confirmed the severe loss of sWAT and ectopic fat deposition in the liver due to the loss of *Agpat2*. Furthermore, they also found an aggravation of hyperlipidemia, liver fibrosis, and atherosclerosis using a double knockout mouse model for the gene encoding the LDL cholesterol receptor (*Ldlr*) (*Agpat2*^−/−^*/Ldlr^−/−^* mice) [107]. In the Nile tilapia (*Oreochromis niloticus)* animal model, transcriptional inhibition of *Agpat2* resulted in abnormal lipid metabolism and oxidative stress in the liver, characterized by vacuolized hepatocytes and increased expression of antioxidant enzymes [108].

CGL is an autosomal recessive disease that causes widespread loss of AT from birth. The affected individuals exhibit accelerated growth, voracious appetite, increased basal energy expenditure, and advanced bone age [24]. Early infantile cardiomyopathy is a specific phenotype for CGL [109]. Diabetes mellitus is also an essential finding of the disease, and it develops mainly during puberty, as it is ketosis resistant and associated with severe islet amyloidosis and beta cell atrophy [110]. Metabolically, about 70% of individuals have hypertriglyceridemia [111,112], and low HDL cholesterol levels are also a critical finding [113,114]. Hyperinsulinemia and increased total cholesterol and LDL-c fraction are metabolic findings of these patients [115,116,117,118]. They also exhibit skeletal muscle hypertrophy, hepatomegaly, and acanthosis nigricans [119,120,121,122,123]. Due to the absence of functional adipocytes, triglycerides are deposited in ectopic tissues such as muscle and the liver [119]. Hepatic steatosis is also common [124]. In *Agpat2* knockout mice, it has been shown that the biosynthesis of diet-derived fat and hepatic triglycerides via a novel monoacylglycerol pathway may contribute to hepatic steatosis [106]. In addition, this same study shows that hepatic fat deposition is not enhanced by *Agpat2* overexpression, suggesting that the role of *AGPAT2* in hepatic lipogenesis is minimal, and fat accumulation in this organ is mainly a consequence of insulin resistance and AT loss. Another study demonstrated that liver fat accumulation in *Agpat2* knockout mice resulted from AT loss and insulin resistance [21,120,125,126,127,128]. Further, Sankella et al. found elevated levels of sphingolipids, such as ceramide C16:0, in the steatotic livers of *Agpat2*^−/−^ mice. These mice also had increased expression levels of enzymes associated with the sphingolipid pathway. These results suggest that ceramide C16:0 could be applied as a biomarker for both insulin resistance and type 2 diabetes mellitus for CGL1 [129].

Female reproductive disorders, such as mild hirsutism, clitoromegaly, oligo-amenorrhea, and polycystic ovaries, were observed in CGL1 women. Most of them are unable to conceive. Reproductive disorders are not observed in men, who usually have normal reproductive capacity [24]. Our research group found that infections and liver diseases are the two leading causes of death in patients with CGL, whose life expectancy for the study population was 27.1 ± 12.4 years [29].

Pathogenic variants in the *AGPAT2* gene are associated with type 1 CGL, which explains the increased focus on studies linked to this gene. In this type, lower serum adiponectin levels are found compared to CGL2. However, leptin levels are higher in CGL2 patients compared to CGL1 [120,130,131,132]. In mice, a deficiency of *Agpat2* impairs insulin signaling and enables unrestricted PA-induced gluconeogenesis, thereby contributing significantly to the development of hyperglycemia [64,133]. In this type, mechanical AT is preserved, which can be explained by increased expression of other AGPAT isoforms or the expression of other genes in these AT depots [134,135,136,137]. In 2009, a case of a patient with CGL who had peripheral hypertonia and reflex excitability was reported. Moreover, magnetic resonance images revealed brain white matter abnormalities, which have been reported in the literature only in this case [138]. Bone cysts and a history of seizures were observed in CGL1 patients [139]. In general, this clinical condition in long bones is more likely in CGL1 individuals compared to CGL2 [140]. Imaging findings demonstrated three types of specific radiographic changes: diffuse osteosclerosis, well-defined osteolytic lesions sparing the axial skeleton, and pseudo-osteopoikilosis in CGL patients, primarily type 1 [141,142]. Another study suggests that in some genetic contexts, *AGPAT2* pathogenic variants can also produce phenotypes with primary polyneuropathy [143]. Specifically, pseudo-osteopoikilosis in the hands and feet has also been reported in a 25-year-old CGL2 woman [144]. Multiple subcutaneous nodules were observed in a 66-day-old CGL1 patient [145]. Regarding bone mineral density (BMD), CGL1 patients have a lower score than CGL2 [146]. In addition, CGL type 1 women have a higher risk of developing diabetes mellitus and acanthosis nigricans than men, the opposite of what occurs for CGL2 [140]. In 2018, a 58-year-old patient of Southeast Asian descent with clinical and metabolic features of familial partial lipodystrophy (FPLD) was diagnosed genetically by whole genome sequencing and had no variants in the FPLD-related gene. However, she presented two heterozygous genetic variants in exons 2 and 4 of the *AGPAT2* gene that had not yet been reported in the databases. In this case, the pathogenic variant in *AGPAT2* seems responsible for an FPLD phenotype rather than CGL. These data suggest the possibility of a polygenic origin for this subtype of FPLD [26].

Along with *AGPAT2*, another gene involved in the etiology of CGL is *BSCL*2, which codes for the protein seipin [127]. Pathogenic variants in this gene are responsible for CGL2 [147]. In this type, a more pronounced fat loss affects both metabolically active and mechanically active AT [134]. Compared to CGL1, CGL2 patients have lower leptin levels, an earlier onset of diabetes, mild cognitive impairment (which may be related to increased seipin expression in the brain [120]), higher insulin levels, and thus insulin resistance [130]. Regarding oxidative stress, our research group identified higher oxidative DNA damage, increased mitochondrial DNA damage, and increased expression of repair enzymes in leukocytes from CGL2 patients compared to heterozygous patients and controls without any pathogenic variant in the *BSCL*2 gene [148]. Our group conducted a systematic review of the muscular aspect of lipodystrophic patients [149]. It raised the presence of muscle impairment in individuals with CGL2, which was not observed in CGL1 patients [150]. However, we have demonstrated that when it comes to maximum respiratory pressure, both CGL1 and 2 individuals had a decrease in this parameter [151].

For a long time, researchers have raised several hypotheses about the molecular pathophysiology of the main types of CGL. The main causes of CGL are impaired lipogenesis (synthesis and storage of triglycerides), blocked adipogenesis (differentiation of preadipocytes into adipocytes), or the apoptosis/necrosis of adipocytes [152]. A study about the pathology of CGL found that *Agpat2* knockout mice died mostly during the first 2 weeks of life. The surviving mice developed severe insulin resistance, hepatic steatosis, and lipoatrophic diabetes [65]. These data corroborate another study that showed *Agpat2* as essential for the postnatal development and maintenance of WAT and BAT. This study showed that the loss of fat depots occurs in the first week of life, precisely the week of high mortality mentioned in the first study. Adipocyte death is caused by autophagy and inflammation [152]. On the contrary, in the same study, massive adipocyte necrosis was found in BAT, demonstrating that the lack of *Agpat2* occurs differently between the two major types of AT. Further studies in *Agpat2^−/−^* mice showed that the loss of both AT stores occurs during the first week of postnatal life due to adipocyte death and inflammatory infiltration of AT. Furthermore, adipocytes from mice lacking *Agpat2* show fewer caveolae, exhibit abnormal mitochondria and LDs, and have abundant autophagic structures [65].

Seipin’s function, in turn, remained unknown for a long time [153]. Initially, it was observed that *BSCL*2 expression was strongly induced during adipocyte differentiation and was also essential for adipogenesis to occur [154]. Further studies in lymphoblastoid cell lines from CGL type 2 patients demonstrated changes in the pattern of LDs that decreased in size and increased in number compared to control cells [155]. It was later found that seipin is a support protein for 1-AGPAT 2 and promotes adipogenesis through the direct inhibition of GPAT3 [92]. Seipin can even directly associate with 1-AGPAT 2 and GPAT3 simultaneously. A study with hepatocarcinoma, osteosarcoma, and lung carcinoma cell lines demonstrated that the knockdown of seipin increases nuclear LDs and PA in the nucleus, while its overexpression decreases these molecules. Thus, it is not directly involved in nuclear LD formation but in its regulation [156]. Molecular studies about the impact of 1-AGPAT 2 protein–protein interactions in AT biology will help us understand how this protein regulates adipogenesis in WAT, BAT, and beige adipocytes in humans.

## 6. Pathogenic Variants in the *AGPAT2* Gene

There is enormous heterogeneity in the pattern of variants found in patients with CGL type 1. A review by Patni et al. [135] showed that more than 90% of CGL1 patients have null pathogenic variants and no detectable enzyme activity in vitro [48]. However, 4% of patients with CGL1 are compound heterozygotes with a null and missense pathogenic variant (with some residual enzyme activity), and only 2% of patients have been reported to have homozygous missense pathogenic variants [21,120,125,126,127]. Although there is a significant difference at the molecular level, this does not seem to affect the expressed clinical symptomatology or the severity of fat loss [135]. We raised the most frequent pathogenic variants in the *AGPAT2* gene. These were c.589-2A>G (11.5%) [21], c.317-588del (7.3%) [21,157], c.202C>T (4.5%) [132,139], c.646A>T (1.2%) [132], c.514G>A (1.2%) [158], and c.144C>A (0.9%) [132], which were relative to the frequency of variants in all four genes causing CGL [159]. The membrane topology predictions for these *AGPAT2* pathogenic variants are provided in Table 1. The software used was PSIPRED (MEMSAT), SOSUI, TMHMM, and T-COFFEE. We observed that there is no consensus among the predictions. For this reason, we chose to demonstrate the predictions of four software programs and their differences and similarities (Table 1). Even in the WT protein, there are distinctions regarding the number of transmembrane domains found. In addition, the N-terminal and C-terminal regions are very distinct among all predictions. The WT protein generally has four transmembrane domains. The T-COFFEE prediction, in turn, showed three domains for 1-AGPAT2. The c.144C>A (p.Cys48*) variant presents only one transmembrane domain according to PSIPRED and SOSUI predictions. In contrast, this variant presents two transmembrane domains, as predicted by TMHMM and T-COFFEE. The c.202 C>T (p.Arg68*) presented a more homogeneous analysis among the software used, showing two domains in all of them, and with most predictions pointing to the N-terminal and C-terminal regions in the cytoplasm. The c.366-588del (p.Leu123Cysfs*56) deletion presented three domains only in the SOSUI analysis. In the other bioinformatics predictions, this variant presents two transmembrane domains, with most predictions pointing the N-terminal and C-terminal regions to the ER lumen. The c.369_372delGCTC (p.Leu124Serfs*26) deletion is generally shown with three transmembrane domains (only TMHMM predicts two domains). The c.514G>A (p.Glu172Lys) variant closely resembles the four transmembrane domains of WT protein that were predicted by PSIPRED, SOSUI, and TMHMM. Only T-COFFEE predicted three transmembrane domains in both WT and p.Glu172Lys. The variants c.570C>A (p.Tyr190*) and c.589-2A>G (p.Val197Alafs*19) present three transmembrane domains according to PSIPRED, TMHMM, and T-COFFEE analyses. However, applying SOSUI analysis, four domains are observed. In the c.646A>T (p.Lys216*) variant, only PSIPED differs from the other software, predicting three transmembrane domains. The N-terminal and C-terminal regions for c.646A>T are mainly expected to be in the ER lumen. The c.369_372delGCTC, c.514G>A, c.570C>A, and c.589-2A>G variants present a significant heterogeneity regarding the prediction of their N- and C-terminal regions.
ijms-26-05416-t001_Table 1Table 1Bioinformatics prediction analysis of the protein sequence of 1-AGPAT 2 and its main mutated proteins.1-AGPAT 2(NP_006403.2)SoftwareTM1TM2TM3TM4N-TerminalC-TerminalWT PSIPRED14–2933–51123–138190–205CytoplasmicER LumenSOSUI2–2430–5258–80122–142ER LumenER LumenTMHMM7–2428–50123–141188–210ER LumenER LumenT-COFFEE31–5059–76123–141-ER LumenCytoplasmicc.144C>APSIPRED15–30---CytoplasmicER LumenSOSUI14–36---ER LumenER LumenTMHMM4–2126–45--CytoplasmicCytoplasmicT-COFFEE5–2130–46--CytoplasmicCytoplasmicc.202C>TPSIPRED15–3036–51--CytoplasmicCytoplasmicSOSUI2–2430–52--ER LumenER LumenTMHMM7–2428–50--ER LumenER LumenT-COFFEE4–2130–50--CytoplasmicCytoplasmicc.366-588delPSIPRED12–2731–50--CytoplasmicCytoplasmicSOSUI2–2430–5258–80-ER LumenER LumenTMHMM7–2428–50--ER LumenER LumenT-COFFEE31–5059–76--ER LumenER Lumenc.369_372delGCTCPSIPRED13–2832–5261–76-CytoplasmicER LumenSOSUI2–2430–5258–80-ER LumenER LumenTMHMM7–2428–50--ER LumenER LumenT-COFFEE4–2130–5059–76-CytoplasmicER Lumenc.514G>APSIPRED13–2832–51123–138190–205CytoplasmicCytoplasmicSOSUI2–2430–5258–80122–142ER LumenER LumenTMHMM7–2428–50123–141188–210ER LumenER LumenT-COFFEE32–5059–76123–141-ER LumenCytoplasmicc.570C>APSIPRED13–2850–32123–138-CytoplasmicER LumenSOSUI2–2430–5258–80122–142ER LumenER LumenTMHMM7–2428–50123–141-ER LumenCytoplasmicT-COFFEE32–5059–76123–141-ER LumenCytoplasmicc.589-2A>GPSIPRED13–2832–50123–138-CytoplasmicER LumenSOSUI2–2430–5258–80122–142ER LumenER LumenTMHMM7–2428–50123–141-ER LumenCytoplasmicT-COFFEE32–5059–76123–141-ER LumenCytoplasmicc.646A>TPSIPRED13–2858–32123–138-CytoplasmicER LumenSOSUI2–2430–5258–80122–142ER LumenER LumenTMHMM7–2428–50123–141188–210ER LumenER LumenT-COFFEE31–5059–76123–141187–207ER LumenER Lumen


It is important to note that to obtain the variant sequence for the 1037-base-pair deletion, we used the gene sequence from the sequence available in NCBI (GRCh38). We manually performed the deletion from the 50th nucleotide of exon 3 to nucleotide +534 in intron 4, according to what was evidenced in the literature, resulting in a protein with 177 amino acids. The commonly used nomenclature for this c.317-588del variant (p.Gly106fs*188) takes into account the deletion from nucleotide +1 of exon 3 to nucleotide +534 of intron 4, resulting in a protein with 188 amino acids, as indicated in the literature in 2002 [21]. Thus, we use the nomenclature c.366-588del (p.Leu123Cysfs*56) for this variant since this nomenclature corresponds to the deletion from the 50th nucleotide of exon 3 to nucleotide +534 in intron 4. The other nucleotide and protein sequences concerning missense variants and the deletion of four base pairs were obtained through the Mutation Taster software (available at https://www.genecascade.org/MutationTaster2021/, accessed on 6 April 2024 [160]. The c.589-2A>G intronic variant has the protein nomenclature described by p.Gln196fs*228, but here, we propose the nomenclature p.Val197Alafs*19, according to HGVS guidelines. Our bioinformatics analyses showed that at position 197, the valine amino acid exchange for an alanine generates a frameshift 17 amino acids later. The glycine shown at position 196 is unchanged in this variant, so we reconsidered its nomenclature (Figure 5).

The c.144C>A pathogenic variant, the fifth most frequent variant in the *AGPAT2* gene, is a cytosine to adenine substitution at position 144. This change generates a premature stop codon and a protein with 47 amino acids. In the c.202C>T variant, the change from thymine to cytosine at position 202 also generates a stop codon, and thus, much of the protein sequence is lost, forming a 67 amino acid product. Deleting 1037 base pairs (c.366-588del) causes the loss of the 50th nucleotide from exon 3 to nucleotide +534 in intron 4. The resulting truncated protein has 177 amino acids. The product of the c.369-372delGCTC deletion has only 148 amino acids [161] (Figure 5).

It is interesting to note that the two analyzed deletions in question occur in very close regions, and although the c.366-588del variant has a deletion of many base pairs, much of what is lost corresponds to introns 3 and 4, leading to an even larger protein than that of the c.369-372delGCTC variant. At the protein level, in c.366-588del, there is a leucine to cysteine change at position 123, which results in a frameshift 56 positions later. While at the following position 124, for the pathogenic variant c.369-372delGCTC, there is a change in the amino acid from leucine to serine, which results in a frameshift of 26 amino acids later (Figure 5). The clinical phenotype of patients with both pathogenic variants is similar, as shown in Table 2.

The substitution of guanine to adenine at position 514 (c.514G>A) [161] does not generate such a significant change in the number of amino acids present in the final protein (278 amino acids). However, in the EGTR domain, which is responsible for the acyltransferase activity of the enzyme, glutamate (E) is exchanged for lysine (K). Glutamate, an amino acid with a negatively charged R group, may be responsible for the incorrect folding of the protein and its impaired catalytic activity when replaced by one with a positively charged R. At position 570, the cytosine to adenine exchange (c.570C>A) results in a protein with 189 amino acids (Figure 5). The most frequent pathogenic variant, c.589-2A>G, is of African origin [120]. This variant affects the splicing acceptor site at position −2 of intron 4, resulting in a mutated protein with 214 amino acids. The c.646A>T variant generates a protein with 215 amino acids.

The variants c.144C>A (p.Cys48*) and c.202C>T (p.Arg68*) give rise to the two smallest proteins analyzed here. The proteins are damaged so early that they lose all four protein motifs in these variants. All other variants analyzed have the NHQSILD domain (NHX4D) preserved. The FINR motif, in turn, remains conserved in the variants c.514G>A (p.Glu172Lys), c.570C>A (p.Tyr190*), 589-2A>G (p.Val197Alafs*19), and c.646A>T (p.Lys216*). Both deletions c.366-588del (p.Leu123Cysfs*56) and c.369_372delGCTC (p.Leu124Serfs*26) lack this protein motif. For the EGTR domain, the variants c.570C>A, 589-2A>G, and c.646A>T present this conserved domain, while in c.514G>A, there is a change from glutamate to lysine in the first amino acid of this domain. The other variants do not present this domain (Figure 5). The fourth and last protein motif described for AGPAT2 (IVPV) is present only in the c.514G>A and c.646A>T variants (Figure 5). Interestingly, the c.646A>T variant conserved all four 1-AGPAT 2 motifs (Figure 5). However, it is a pathogenic variant. Therefore, the decrease in the number of amino acids combined with changes in their polarity is responsible for altering protein folding and, consequently, its correct activity.
ijms-26-05416-t002_Table 2Table 2Genotype–phenotype characteristics of the most frequent pathogenic variants in the *AGPAT2* gene.Pathogenic Variant(NM_006412.4)c.144C>Ac.202C>Tc.366_588delc.369_372delGCTCc.514G>Ac.570C>Ac.589-2A>Gc.646A>TReferences [132][132,139][157][162][158][126][21][132]Resulting 1-AGPAT 2 protein (NP_006403.2)p.Cys48*p.Arg68*p.Leu123Cysfs*56p.Leu124Serfs*26p.Glu172Lysp.Tyr190*p.Val197Alafs*19p.Lys216*Protein consequenceSmaller and truncated protein with 47 aaSmaller and truncated protein with 67 aaSmaller and truncated protein with 177 aaSmaller and truncated protein with 148 aaPoorly functional protein with 278 aaSmaller and truncated protein with 189 aaSmaller and truncated protein with 214 aaSmaller and truncated protein with 215 aaProtein domains affectedAll domains absentAll domains absentEGTR, FINR and IVPV domains absentEGTR, FINR and IVPV domains absentEGTR domain affected: change from E (glutamate) to K (lysine) aaIVPV domain absentIVPV domain absentAll domains preservedNumber of patients (n)32 [132]; 2 [139]1022151Age (average in years)2813 [132]; 63 [139]4080,3201925Generalized lack of subcutaneous WAT (sWAT)++ [132]; + [139]++++++Hypertriglyceridemia++ [132]; - [139]++++++Diabetes mellitus 2+- [132]; + [139]++++++++Acanthosis nigricans+- [132]; + [139]++-+-+Insulin resistance-+++ [132]; - [139]+++-+-+++-Retinopathy+- [132]; + [139]-----+Diabetic neuropathy+- [132]; - [139]-----+Recurrent acute pancreatitis+- [132]; - [139]------Splenic artery aneurysm+- [132]; - [139]------Hepatomegaly-- [132]; - [139]+++---Bone cysts+- [132]; + [139]--+---Polycystic ovary+- [132]; - [139]--+---Hypertension+- [132]; + [139]--+---Renal failure+- [132]^;^ - [139]------Muscular hypertrophy-- [132]; - [139]++++--Inguinal hernia-+ [132]; - [139]------Umbilical hernia-- [132]; - [139]-+----Increased abdominal volume-- [132]; - [139]-+----Hepatic steatosis-- [132]; - [139]------Large ears-- [132]; - [139]--+---Genital dysmorphism-+ [132]; - [139]--+---Acromegaloid dysmorphism++ [132]; - [139]++-+---Hirsutism-- [132]; - [139]+-----The pathogenic variants that generated more severe phenotypic characteristics are related to alterations or the absence of the EGTR domain, which is responsible for the acyltransferase activity of 1-AGPAT 2. The aggressiveness of the phenotype was demonstrated as +: present; ++: median; +++: severe. Unreported phenotypes are indicated by -.


The genotype–phenotype heterogeneity observed in CGL individuals is clinically important. However, there is a limited correlation between the most prevalent *AGPAT2* pathogenic variants described in the literature, their possible effect on the protein level, and their main metabolic commitments [162]. This review also contributes to unraveling these questions. Table 1 highlights that CGL1 patients harboring pathogenic variants that did not affect the EGTR and IVPV 1-AGPAT 2 domains presented more CGL1 clinical signs and symptoms than those with pathogenic variants affecting at least one of the domains. In addition, we observed that the absence of the EGTR domain per se is related to a more aggressive disease phenotype. Finally, the pathogenic variants analyzed that exclude the EGTR and IVPV 1-AGPAT 2 domains are related to a worse clinical CGL1 presentation. Thus, depicting the *AGPAT2* molecular heterogeneity can help the diagnosis and clinical management of CGL1.

## 7. Concluding Remarks and Future Directions

This review highlighted several findings concerning the role of the *AGPAT2* gene on adipogenesis and its relationship with CGL1 development. We also provided a genotype–phenotype correlation to better understand the CGL1 phenotypic heterogeneity. Multiple sequence alignments of human wild-type and mutated 1-AGPAT 2 performed here showed that different *AGPAT2* pathogenic variants can result in different protein changes, affecting different domains of 1-AGPAT 2. Genotype–phenotype correlation data can also help clinicians understand the degree of metabolic commitments of CGL1. However, many questions remain to be investigated to improve our knowledge in this field. It is crucial to scrutinize the role of the *AGPAT2* gene in WAT, BAT, and beige adipogenesis, which could give us new insights into its functions in health and disease.

## Figures and Tables

**Figure 1 ijms-26-05416-f001:**
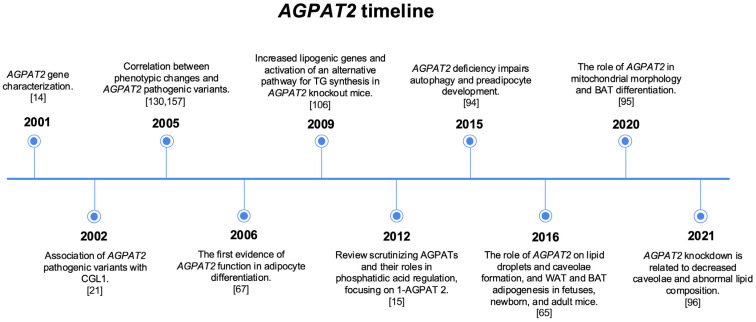
This timeline highlights the discovery, characterization, mechanisms of action, and role of the *AGPAT2* gene in adipogenesis and CGL1 development.

**Figure 2 ijms-26-05416-f002:**
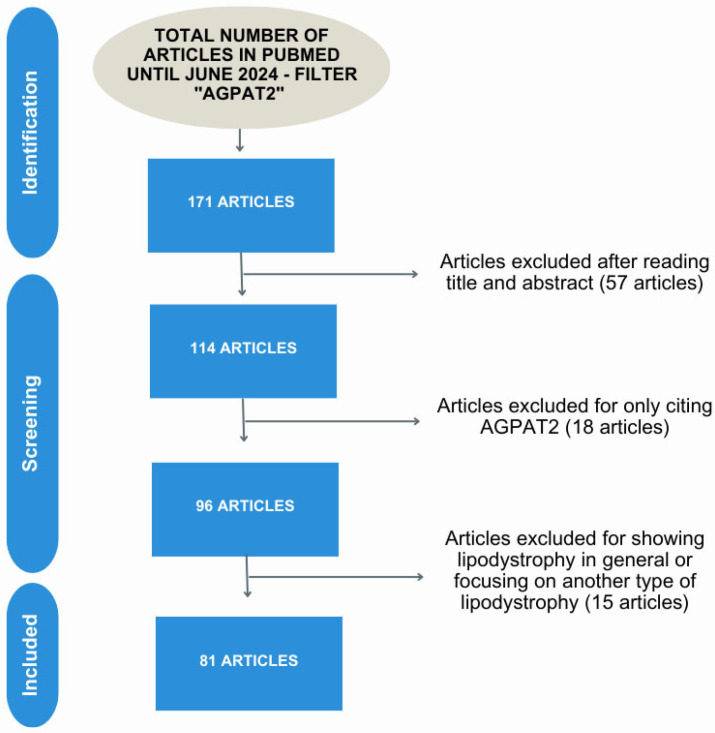
A representative flowchart of the studies targeted in this review. The screening of the studies was performed based on three exclusion criteria: 57 articles were excluded after reading the title and abstract for not having the focus of the survey on *AGPAT2*-related diseases and not having *AGPAT2* as an essential target in the research, only citing it (18 articles excluded), and 15 articles were excluded for dealing with another type of lipodystrophy or citing the disease only in a general way. Thus, 81 articles directly related to the *AGPAT2* gene were included in this review.

**Figure 3 ijms-26-05416-f003:**
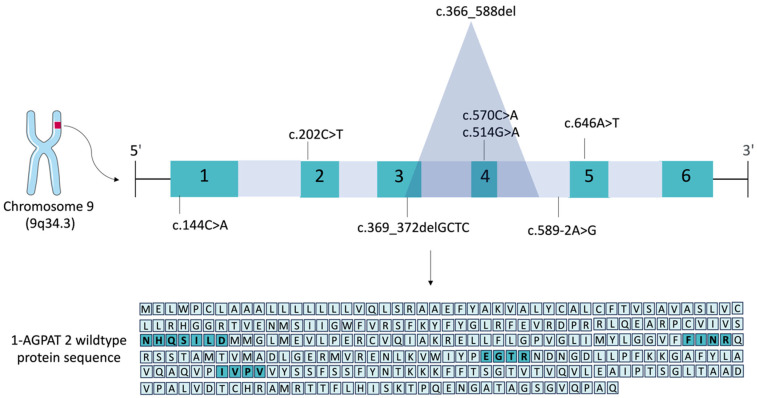
The main *AGPAT2* pathogenic variants related to CGL1. 1-AGPAT 2 is encoded from the *AGPAT2* gene located on the short arm of chromosome 9 at position 34.2. This gene has six exons and five introns; most of the main pathogenic variants already reported are located in coding regions, although some intronic variants also exist. Wild-type 1-AGPAT 2 has four major domains: NHQSILD, FINR, EGTR, and IVPV. The nomenclature of all *AGPAT2* pathogenic variants was based on the transcript NM_006412.4; ENST00000371696.7. The wild-type 1-AGPAT 2 protein has 278 amino acids (NP_006403.2).

**Figure 4 ijms-26-05416-f004:**
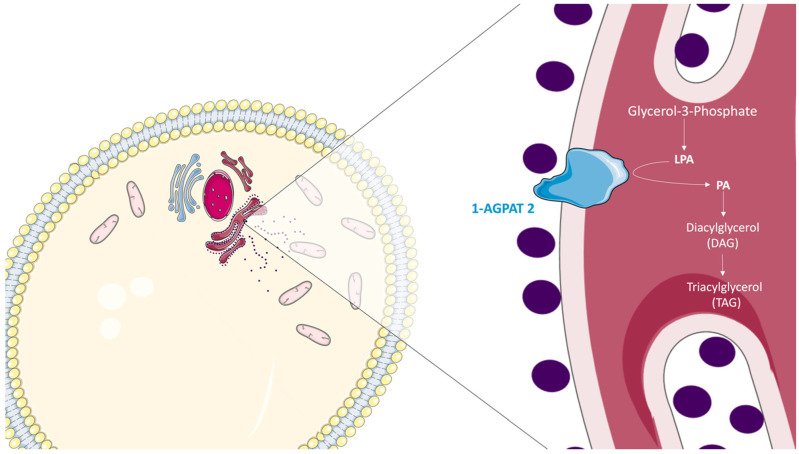
The role of 1-AGPAT 2 in the biosynthesis of triacylglycerols. In adipose tissue (AT), glycerol-3-phosphate is converted into LPA. LPA then becomes a substrate for 1-AGPAT2, which is located in the ER lumen. 1-AGPAT 2 esterifies the sn-2 carbon of phosphorylated 1-acylglycerol on the sn-3 carbon, forming PA. The PA resulting from this reaction will be used to form DAGs and TAGs. TAGs can be stored in AT or participate in the formation of membrane PLs. LPA: lysophosphatidic acid; PA: phosphatidic acid (1,2-diacylglycerol-3-phosphate).

**Figure 5 ijms-26-05416-f005:**
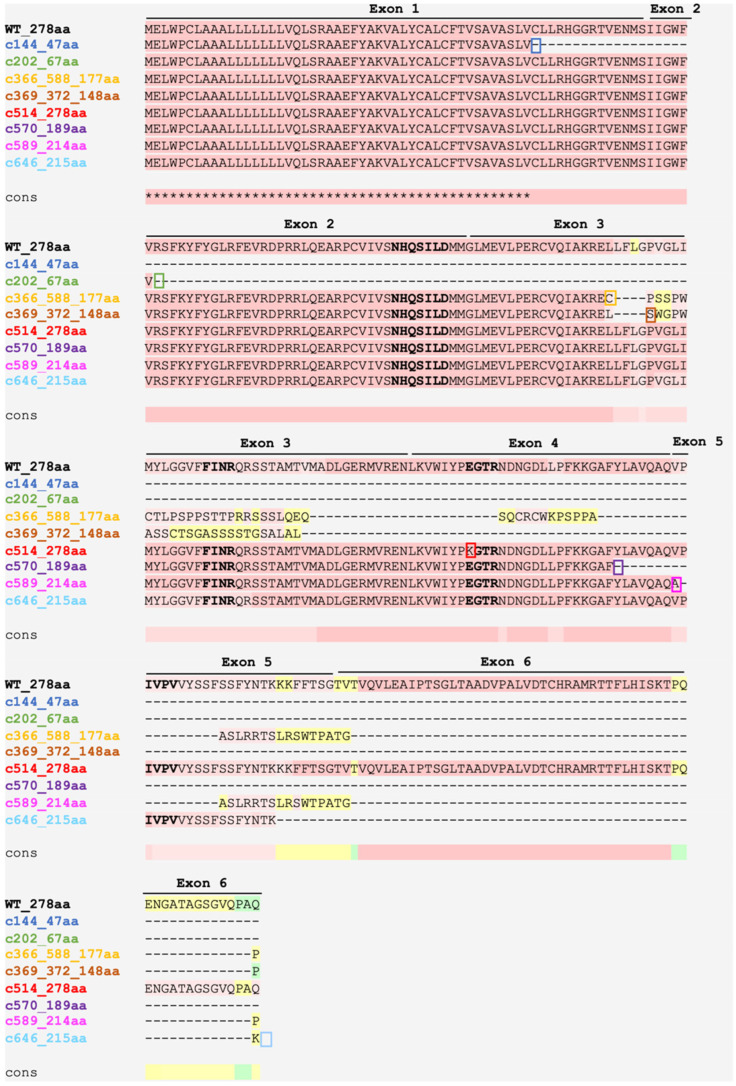
Amino acids sequence alignments of WT and mutated 1-AGPAT 2 proteins. Amino acids highlighted in pink indicate an excellent alignment between the 1-AGPAT 2 amino acid sequences. The yellow color indicates an average alignment, and the green color indicates a poor alignment. ∗ corresponds to an equal match. Cons: consensus sequence. The position of the stop codon in the variants c.144C>A (highlighted in blue), c.202C>T (highlighted in green), c.570C>A (highlighted in purple), and c.646A>T (highlighted in cyan) is shown by blue, green, purple and cyan color boxes, respectively. The functional domains are in bold, and it is possible to observe the lack of all domains in the c.144C>A and c.202C>T variants. The NHQSILD domain is present in all other variants. The FINR motif is present only in variants c.514G>A (highlighted in red), c.570C>A (highlighted in purple), 589-2A>G (highlighted in magenta), and c.646A>T (highlighted in cyan). EGTR is present and unchanged in c.570C>A, 589-2A>G, and c.646A>T, and there is a change from glutamate to lysine in the first amino acid of this domain in the c.514G>A variant (surrounded in a red box). IVPV is present only in the WT and c.514G>A and c.646A>T variants. The position of the mutated amino acid in the variants c.366_588del (highlighted in yellow), c.369_372delGCTC (highlighted in orange), and c.589-2A>G (highlighted in magenta) is shown by yellow, orange, and magenta colors boxes, respectively. The 1-AGPAT 2 sequence used was NP_006403.2.

## Data Availability

The original contributions presented in the study are included in the article; further inquiries can be directed to the corresponding authors.

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
