# Peer review of "The Role of the AGPAT2 Gene in Adipose Tissue Biology and Congenital Generalized Lipodystrophy Pathophysiology"

_ijms, 2025, doi:10.3390/ijms26115416_

Round 1
Reviewer 1 Report
Comments and Suggestions for Authors
The authors describe 1-Acylglycerol-3-phosphate O-acyltransferase (1-AGPAT2) and its role in adipose tissue and its importance in adipogenesis. They report the most frequent variants of this gene that lead to congenital generalized lipodystrophy type 1 (CGL1), a condition marked by the congenital absence of adipose tissue and severe metabolic disturbances. The most innovative aspect of this work is the use of bioinformatics tools to assess the effects of key AGPAT2 mutations on 1-AGPAT2 membrane topology and the investigation of the genotype-phenotype correlation in CGL1 patients.
The review is well written and offers a comprehensive overview, clearly reflecting the significant effort and thorough research carried out by the authors.
However, I have the following comments:
Major comments
1. The current title suggests that the review provides a detailed discussion on the different types of adipose tissue (white, brown, and pink) and their various sources (subcutaneous, visceral, omental, epicardial, etc.). However, after a discussion about the role of AGPAT2 in the adipose tissue and in role in adipogenesis, the actual focus of the review is on CGL. More importantly, and in my opinion, the key strength of this work is the proposed genotype-phenotype correlation between the most frequent variants and the observed phenotype. Therefore, I recommend modifying the title to immediately highlight the core focus of the review.
2. The text provides detailed information on conserved motifs within the AGPAT family and mentions the presence of transmembrane domains. However, it does not specify exactly which transmembrane domains contain these motifs. Clarifying this would strengthen the discussion on protein structure and function.
3. Figure 3 currently represents the wild-type AGPAT2 isoform. Would it be possible to add a structural schematic of the protein, indicating the four transmembrane domains and where the 4 domains discussed are localized? This would make it easier to visualize the membrane topology.
Minor comments
1. In lines 37-38 and 125-126 (and others), the full term should be introduced before using the abbreviation for the first time to enhance clarity.
2. Line 98 – AGPAT2 Pathogenic Variants: The phrase “the main AGPAT2 pathogenic variants” should specify the source of classification. Are these variants the most frequent or the most deleterious? This is later explained in more detail, but it would be beneficial to provide a clearer description earlier in the text.
3. In line 258, the phrase “Gale et al. 2006 revealed” should be corrected to “Gale et al. in 2006 revealed.”
4. When introducing the CGL1, please consider including information on OMIM classification and prevalence data to provide a broader clinical context.
Author Response
The authors describe 1-Acylglycerol-3-phosphate O-acyltransferase (1-AGPAT2) and its role in adipose tissue and its importance in adipogenesis. They report the most frequent variants of this gene that lead to congenital generalized lipodystrophy type 1 (CGL1), a condition marked by the congenital absence of adipose tissue and severe metabolic disturbances. The most innovative aspect of this work is the use of bioinformatics tools to assess the effects of key AGPAT2 mutations on 1-AGPAT2 membrane topology and the investigation of the genotype-phenotype correlation in CGL1 patients.
The review is well written and offers a comprehensive overview, clearly reflecting the significant effort and thorough research carried out by the authors.
However, I have the following comments:
Major comments
- The current title suggests that the review provides a detailed discussion on the different types of adipose tissue (white, brown, and pink) and their various sources (subcutaneous, visceral, omental, epicardial, etc.). However, after a discussion about the role of AGPAT2 in the adipose tissue and in role in adipogenesis, the actual focus of the review is on CGL. More importantly, and in my opinion, the key strength of this work is the proposed genotype-phenotype correlation between the most frequent variants and the observed phenotype. Therefore, I recommend modifying the title to immediately highlight the core focus of the review.
Answer: Thank you very much for your suggestion. We have rewritten the title: Role of the AGPAT2 Gene in Adipose Tissue Biology and Congenital Generalized Lipodystrophy Pathophysiology.
- The text provides detailed information on conserved motifs within the AGPAT family and mentions the presence of transmembrane domains. However, it does not specify exactly which transmembrane domains contain these motifs. Clarifying this would strengthen the discussion on protein structure and function.
Answer: Thank you very much for your observation. We agree with you, but as shown in Table 1, we applied distinct bioinformatics prediction analyses of the protein sequence of 1-AGPAT 2 and its main variants, and some discrepancies were found about the 1-AGPAT 2 membrane topology. So, we decided to put these observations in Table 1 and discuss them in the text. Further, the four major domains NHX4D, FINR, EGTR, and IVPV were highlighted in Figures 3 and 5, helping to understand how the main AGPAT2 pathogenic variants affect 1-AGPAT 2 functions.
- Figure 3 currently represents the wild-type AGPAT2 isoform. Would it be possible to add a structural schematic of the protein, indicating the four transmembrane domains and where the 4 domains discussed are localized? This would make it easier to visualize the membrane topology.
Answer: Thank you very much for your observation. As we informed in the manuscript (lines 550-551): “Regarding the membrane topologies, it is observed that there is no consensus. For this reason, we chose to demonstrate the prediction of 4 software programs and their differences and similarities”. For this purpose, as we informed above, we decided to put these observations in Table 1 and discuss them in the text.
Minor comments
- In lines 37-38 and 125-126 (and others), the full term should be introduced before using the abbreviation for the first time to enhance clarity.
Answer: Thank you very much for your observation. We have made the requested changes to better understand the terms in the article.
2. Line 98 – AGPAT2 Pathogenic Variants: The phrase “the main AGPAT2 pathogenic variants” should specify the source of classification. Are these variants the most frequent or the most deleterious? This is later explained in more detail, but it would be beneficial to provide a clearer description earlier in the text.
Answer: Thank you very much for your observation. We included the most frequent pathogenic variants. We changed the text (lines 102-103).
3. In line 258, the phrase “Gale et al. 2006 revealed” should be corrected to “Gale et al. in 2006 revealed.”
Answer: Thank you very much for your observation. We have made the necessary changes to standardize the citation format throughout the article.
4. When introducing the CGL1, please consider including information on OMIM classification and prevalence data to provide a broader clinical context.
Answer: Thank you very much for your suggestion. We added the OMIM classification for CGL1 (line 47) and its prevalence (lines 62-63).
Reviewer 2 Report
Comments and Suggestions for Authors
The manuscript ijms-3555707 summarizes the effect of AGPAT2 gene variants on the pathogenesis, biochemistry, and metabolism of CGL1. This manuscript is well-written and comprehensive; however, minor revision would be helpful to increase readability.
- Please revise Figure 4 (e.g. adding arrows to show increase or decrease the metabolic response) so that the effects of AGPAT2 gene variants on the biosynthesis of triacylglycerols are shown.
- Table 2 is very helpful to check the phenotype induced by AGPAT2 gene variants. I wonder if the authors could add other variables, such as blood lipid profiles.
Author Response
The manuscript ijms-3555707 summarizes the effect of AGPAT2 gene variants on the pathogenesis, biochemistry, and metabolism of CGL1. This manuscript is well-written and comprehensive; however, minor revision would be helpful to increase readability.
- Please revise Figure 4 (e.g. adding arrows to show increase or decrease the metabolic response) so that the effects of AGPAT2 gene variants on the biosynthesis of triacylglycerols are shown.
Answer: Thank you very much for your suggestion. However, we did not measure the activity of all mutated 1-AGPATs 2 reviewed here. So, we cannot discuss their effects on the biosynthesis of triacylglycerols (TGs). Further, other AGPAT isoforms and other pathways can do the same reaction as 1 AGPAT-2. The goal of this image was to highlight the role of 1-AGPAT2 in TG biosynthesis. We discussed the effects of overexpressing or downregulating AGPAT2 in different cell models to clarify its role in TG biosynthesis.
2. Table 2 is very helpful to check the phenotype induced by AGPAT2 gene variants. I wonder if the authors could add other variables, such as blood lipid profiles.
Answer: Thank you very much for your suggestion. However, the main biochemical finding in reviewed papers was the level of triacylglycerols (TGs). Diabetes and insulin resistance were also included. We also discussed blood lipid profiles in the main text (lines 363-366).
Reviewer 3 Report
Comments and Suggestions for Authors
Manuscript ID: ijms-3555707
Title: Role of the AGPAT2 gene in adipose tissue biology
In this study, the authors comprehensively reviewed the discovery of the AGPAT2 gene, elucidated its protein expression patterns and post-translational modifications, and systematically summarized its functional implications in lipid metabolism. Furthermore, bioinformatics approaches were employed to analyze structural and functional alterations associated with AGPAT2 mutations. It’s a very interesting study. Only few questions should be addressed to make it optimized.
- Lines 42-43: “This enzyme is highlighted by its remarkable and essential role in providing substrates for the synthesis of TAG and GPL.” I disagree with the description, 1-AGPAT 2 the substrates of TAG and GPL synthesis excluding 1-AGPAT 2, please check it.
- Lines 256-309: A literature review requires critical synthesis and analytical integration of existing studies rather than mere cataloging of findings. The manuscript should be comprehensively restructured to emphasize thematic connections and conceptual progression throughout.
- Lines 883: Reference #158 exhibits non-compliance with the required referencing format, please check it.
Author Response
In this study, the authors comprehensively reviewed the discovery of the AGPAT2 gene, elucidated its protein expression patterns and post-translational modifications, and systematically summarized its functional implications in lipid metabolism. Furthermore, bioinformatics approaches were employed to analyze structural and functional alterations associated with AGPAT2 mutations. It’s a very interesting study. Only few questions should be addressed to make it optimized.
- Lines 42-43: “This enzyme is highlighted by its remarkable and essential role in providing substrates for the synthesis of TAG and GPL.” I disagree with the description, 1-AGPAT 2 the substrates of TAG and GPL synthesis excluding 1-AGPAT 2, please check it.
Answer: Thank you very much for your suggestion. We checked this information, and it is correct. To clarify, we made a little adjustment to this sentence (line 46).
2. Lines 256-309: A literature review requires critical synthesis and analytical integration of existing studies rather than mere cataloging of findings. The manuscript should be comprehensively restructured to emphasize thematic connections and conceptual progression throughout.
Answer: Thank you very much for your suggestion. We improved the connections in the manuscript, and we hope it is now appropriate for publication.
3. Lines 883: Reference #158 exhibits non-compliance with the required referencing format, please check it.
Answer: Thank you very much for this observation. We corrected this reference, which is now the reference 162.
Reviewer 4 Report
Comments and Suggestions for Authors
Manuscript titled “Role of the AGPAT2 Gene in Adipose Tissue Biology” provides a review about AGPAT2 and its role on lipid metabolism, specifically, some disease-associated variants are also considered. There is interesting information reported in the document; there are some comments and suggestions for the authors:
- In the abstract, please consider providing a brief conclusion, since it currently ends rather abruptly.
- Keywords “AGPAT2” and “adipose tissue” are already present on your title. Please consider substituting these keywords for others not present in the title, in order to make your published paper easier to discover.
- The introduction describes various molecular features of the AGPAT2, as well as highlighting its role in CGL1. In addition to this information, please consider providing data regarding the clinical aspects of the disease, such as incidence, treatment, life expectancy of the patients, current treatments, changes to quality of life, etc. Providing these or any other information that the authors consider appropriate will make your document more relevant.
- The legend of figure 4 describes the reactions depicted in the image, although only the names of the molecules are actually presented. Is it possible to include the chemical structures to visualize the process in greater detail? The precise enzyme mechanism would also be useful to include if available and if possible. Another suggestion (again, if the authors consider it appropriate) is to specify what molecules are accumulated and/or deficient in the absence of the enzyme’s activity, and which organs/cells/enzymes/etc. they may negatively affect, thereby resulting in negative health effects. Some of this information is mentioned in the main text, but visualizing it in the image could make the document more understandable and informative.
- Line 251 mentions “PPARG”, while line 274 mentions “PPARγ” and its definition. It appears that “PPARG” and “PPARγ” are actually the same molecule, if so, please homogenize their names and define the abbreviation when it is first mentioned.
- Line 375 mentions that “mechanical AT is preserved”. If known, please specify which specific adipose deposits are preserved. Likewise, please specify the tissues where “higher oxidative DNA damage, increased mitochondrial DNA damage, and increased expression of repair enzymes” was documented (lines 404-405).
- Table 2 provides an interesting summary of genotype-phenotype associations. Are these variants listed in any specific order? Please mention it if this is the case.
Author Response
Reviewer 4
Manuscript titled “Role of the AGPAT2 Gene in Adipose Tissue Biology” provides a review about AGPAT2 and its role on lipid metabolism, specifically, some disease-associated variants are also considered. There is interesting information reported in the document; there are some comments and suggestions for the authors:
- In the abstract, please consider providing a brief conclusion, since it currently ends rather abruptly.
Answer: Thank you very much for your observation. We made some modifications to the abstract.
2. Keywords “AGPAT2” and “adipose tissue” are already present on your title. Please consider substituting these keywords for others not present in the title, in order to make your published paper easier to discover.
Answer: Thank you very much for your observation. We changed the keywords in order to facilitate the discovery of this article.
3. The introduction describes various molecular features of the AGPAT2, as well as highlighting its role in CGL1. In addition to this information, please consider providing data regarding the clinical aspects of the disease, such as incidence, treatment, life expectancy of the patients, current treatments, changes to quality of life, etc. Providing these or any other information that the authors consider appropriate will make your document more relevant.
Answer: We appreciate your observation. As informed to reviewer 1, we added CGL prevalence data (lines 62-63). The other data was included in the same paragraph (lines 63-66).
4. The legend of figure 4 describes the reactions depicted in the image, although only the names of the molecules are actually presented. Is it possible to include the chemical structures to visualize the process in greater detail? The precise enzyme mechanism would also be useful to include if available and if possible. Another suggestion (again, if the authors consider it appropriate) is to specify what molecules are accumulated and/or deficient in the absence of the enzyme’s activity, and which organs/cells/enzymes/etc. they may negatively affect, thereby resulting in negative health effects. Some of this information is mentioned in the main text, but visualizing it in the image could make the document more understandable and informative.
Answer: Thank you very much for your suggestion. However, as informed to reviewer 2, we did not measure the activity of all mutated 1-AGPATs 2 reviewed here. So, we cannot discuss their effects on the biosynthesis of triacylglycerols (TGs). Further, other AGPAT isoforms and other pathways can do the same reaction as 1 AGPAT-2. The goal of this image was to highlight the role of 1-AGPAT2 in TG biosynthesis. We discussed the effects of overexpressing or downregulating AGPAT2 in different cell models to clarify its role in TG biosynthesis.
5. Line 251 mentions “PPARG”, while line 274 mentions “PPARγ” and its definition. It appears that “PPARG” and “PPARγ” are actually the same molecule, if so, please homogenize their names and define the abbreviation when it is first mentioned.
Answer: Thank you very much for your observation. We modify (line 261 and 285) the term for PPARγ.
6. Line 375 mentions that “mechanical AT is preserved”. If known, please specify which specific adipose deposits are preserved. Likewise, please specify the tissues where “higher oxidative DNA damage, increased mitochondrial DNA damage, and increased expression of repair enzymes” was documented (lines 404-405).
Answer: We appreciate your observation. Regarding the DNA damage, we included (line 420) the referent cells (leukocytes) utilized for the analysis and inserted the cell model in the text.
7. Table 2 provides an interesting summary of genotype-phenotype associations. Are these variants listed in any specific order? Please mention it if this is the case.
Answer: Thank you very much for your observation. The order of variants in Table 2 concerns their appearance in the AGPAT2 gene.
Round 2
Reviewer 4 Report
Comments and Suggestions for Authors
Manuscript titled “Role of the AGPAT2 Gene in Adipose Tissue Biology and Congenital Generalized Lipodystrophy Pathophysiology” provides a review about AGPAT2 and its role on lipid metabolism. The present version of the manuscript was modified according to comments and suggestions made during an initial revision; those made by the present reviewer include:
- Adding a brief concluding remark to the abstract in order to avoid an abrupt ending. A brief conclusion was added to the abstract.
- Avoiding keywords that were already present on the manuscript’s title, in order to increase the discoverability of the published paper. Repeated keywords were replaced.
- Complementing the introduction with some relevant clinical aspects of the disease. Additional information was added.
- Adding additional information to figure 4. The authors comment that they do not have such information available to add.
- Homogenizing the terms “PPARG” “PPARγ”. They were homogenized.
- Confirming where possible, the adipose tissues preserved, as well as oxidative DNA damage, as mentioned in the original document. The authors have specified DNA damage on leukocytes.
- Confirming if the genotype-phenotype associations listed in Table 2 are sorted in any specific order. The authors confirm that they are mentioned as they appear in the gene.
According to the aforementioned changes made by the authors, it is apparent that they adequately considered and addressed all comments and suggestions made by the present reviewer. There are no additional ones to suggest for the present version of the document.